# The Extracellular Matrix and Cardiac Pressure Overload: Focus on Novel Treatment Targets

**DOI:** 10.3390/cells13201685

**Published:** 2024-10-12

**Authors:** Matthijs Snelders, Meltem Yildirim, A. H. Jan Danser, Ingrid van der Pluijm, Jeroen Essers

**Affiliations:** 1Department of Molecular Genetics, Erasmus University Medical Center, 3015 GD Rotterdam, The Netherlands; 2Division of Pharmacology, Department of Internal Medicine, Erasmus University Medical Center, 3015 GD Rotterdam, The Netherlands; a.danser@erasmusmc.nl; 3Department of Vascular Surgery, Cardiovascular Institute, Erasmus University Medical Center, 3015 GD Rotterdam, The Netherlands; 4Department of Radiotherapy, Erasmus University Medical Center, 3015 GD Rotterdam, The Netherlands

**Keywords:** heart failure, cardiac remodeling, intervention, extracellular matrix, renin angiotensin system

## Abstract

Heart failure is a significant health issue in developed countries, often stemming from conditions like hypertension, which imposes a pressure overload on the heart. Despite various treatment strategies for heart failure, many lack long-term effectiveness. A critical aspect of cardiac disease is the remodeling of the heart, where compensatory changes in the extracellular matrix exacerbate disease progression. This review explores the processes and changes occurring in the pressure-overloaded heart with respect to the extracellular matrix. It further summarizes current treatment strategies, and then focuses on novel treatment targets for maladaptive cardiac remodeling, derived from transverse aortic constriction-induced pressure overload animal models.

## 1. Introduction to Cardiac Pressure Overload

Cardiac pressure overload, often caused by hypertension, significantly contributes to mortality in developed nations. It is characterized by cardiac remodeling resulting in fibrosis and by left ventricular hypertrophy. Recent therapeutic approaches aim to mitigate pressure overload and its associated effects, particularly by targeting the renin-angiotensin system (RAS), which is pivotal in the regulation of blood pressure, sodium balance, and cardiac and vascular remodeling [1]. Dysregulation of the RAS profoundly affects vascular homeostasis and cardiac function. Other treatment options include mineralocorticoid receptor antagonists (MRAs), sodium-glucose cotransporter 2 inhibitors (SGLT2i), and beta-adrenergic receptor blockers. While treatment aimed at lowering blood pressure may initially alleviate acute effects, chronic pressure overload leads to maladaptive remodeling of the left ventricle and myocardium. Therefore, optimized treatment is required to not only halt disease progression, but also to reverse it. Ongoing research investigates novel therapies targeting pathological cardiac remodeling. This review discusses the challenges in managing chronic pressure overload, provides an overview of current therapies for treating cardiac pressure overload, and explores the ECM as a therapeutic target by summarizing existing and potential future therapies investigated in animal models of cardiac pressure overload.

## 2. Challenges in Halting Disease Progression of the Pressure-Overloaded Heart

Initially, cardiac pressure overload, when discovered early, can be managed through normalizing blood pressure. Apart from the acute effects of hypertension, the local effects of angiotensin (Ang) II, aldosterone, and noradrenaline may play a role in disease progression. Long-term effects of pressure overload include maladaptive morphological changes that eventually reduce treatment effectiveness. Combination therapy may offer greater efficacy than single-drug treatments [2], but care should be taken to avoid the increased risk of side effects. Understanding ECM protein homeostasis is crucial to elucidate pathophysiological mechanisms involved in cardiac remodeling following pressure overload-induced injury.

### 2.1. Cardiac Remodeling in Response to Pressure Overload

Cardiac remodeling is defined as an altered gene expression in response to wall stress, leading to molecular, cellular, and interstitial changes. It is manifested clinically as changes in heart size, shape, and function [3]. The ECM, comprising fibrillar and non-fibrillar proteins, plays critical roles in heart structure and signaling and is altered in response to chronic pressure changes [4,5]. ECM proteins are synthesized and degraded in response to mechanical cues or cellular signaling. Molecular responses like Ang II binding to Ang II type 1 receptors (AT1R) on vascular smooth muscle cells, triggering transforming growth factor β1 (TGFβ1) production, illustrate the interplay of mechanical stress from pressure overload and increased TGFβ1 levels in promoting maladaptive ECM remodeling (Figure 1).

The cellular composition of mammalian cardiac tissue is well-documented, with prominent cell types including cardiomyocytes (23%), endothelial cells (29%), and fibroblasts (25%) [6]. Cardiac fibroblasts play pivotal roles in the hypertrophic response of cardiomyocytes during pressure overload in mice [7]. Initially adaptive, cardiomyocyte hypertrophy increases cell size to manage increased cardiac workload [8]. Prolonged hypertrophy transitions to maladaptive remodeling, characterized by cardiac injury, fibroblast activation, and expansion at sites of pressure overload, such as the left atrium, septum, and left ventricular wall [9,10]. Within five days of pressure overload, the cardiac fibroblast population expands, and fibroblasts transdifferentiate to myofibroblasts, increasing their expression of α-smooth muscle cell actin (α-SMA) and stimulating collagen deposition [11]. Progressive cardiomyocyte loss and myofibroblast transdifferentiation from activated fibroblasts contribute to increased collagen deposition, fibrotic scar tissue development, systolic and diastolic dysfunction, and eventual heart failure [12,13].

Clinically, the detection of fibrotic remodeling is significantly enhanced through myocardial strain imaging, specifically using speckle-tracking echocardiography (STE). This innovative imaging modality allows cardiologists to identify subclinical impairments in myocardial strain parameters before the onset of overt left ventricular systolic dysfunction, typically indicated by a decline in the left ventricular ejection fraction. By quantifying the left ventricular global longitudinal strain (LV-GLS), STE offers valuable insights into the myocardial condition. A lower magnitude of LV-GLS is strongly associated with greater collagen deposition in myocardial tissue [14], reflecting the extent of myocardial fibrosis. Importantly, LV-GLS has been shown to correlate well with histological assessments of fibrosis [15,16], making it a critical tool for early detection and monitoring of cardiac remodeling. Moreover, unlike traditional techniques such as cardiac magnetic resonance imaging, which often requires contrast agents like gadolinium, STE is entirely non-invasive and does not involve exposure to any contrast materials. This makes STE particularly advantageous in clinical practice for timely intervention in patients at risk of heart failure due to myocardial fibrosis.

Understanding ECM remodeling mechanisms in cardiac pressure overload often provides insight into similar processes in aortic diseases. For example, Marfan syndrome is caused by a defect in an ECM component found in both the heart and aorta. Since there is a correlation between extracellular matrix (ECM) changes in pressure overload and aortic diseases like those observed in Marfan syndrome [17,18], insights into ECM remodeling in aortic pathologies can additionally inform us on novel approaches to heart failure.

### 2.2. Mouse Models to Study Cardiac Pressure Overload

Current strategies to study cardiac pressure overload involve the use of mouse models. One commonly used model to study cardiac remodeling after cardiac pressure overload is the use of transverse aortic constriction (TAC). As the name suggests, TAC mimics pressure overload by reducing the diameter of the mouse aorta using a 25 to 27 gauge needle [19]. This significant constriction results in an up to 70% reduction of left ventricular outflow, depending on the resulting diameter, leading to cardiac pressure overload and subsequent left ventricular hypertrophy and, eventually, heart failure [20]. This allows researchers to follow the progression of left ventricular maladaptive hypertrophy to heart failure, keeping in mind that the site of constriction (before or after the kidney, with only the former resulting in renal hypoperfusion) determines whether the RAS is simultaneously activated [19,21]. Mice with genetic defects related to connective tissue disorders, such as Marfan syndrome [22], can also be used in conjunction with TAC to further explore the effect on the ECM during cardiac pressure overload.

## 3. Current Therapies for Cardiac Pressure Overload

Preclinical studies have helped to reveal which hormonal systems regulate blood pressure and the cardiac adaptive response in heart failure, and based on this, several drug classes are now being used to treat this disorder.

### 3.1. Inhibition of the Renin-Angiotensin System: Renin Inhibitors, Angiotensin-Converting Enzyme Inhibitors, and Angiotensin II Type 1 Receptor Blockers

The RAS is one of the main systems that regulates blood pressure. Ang II, converted from Ang I by angiotensin-converting enzymes (ACEs), plays a key role in the RAS. Ang II activates AT1R and Ang II type 2 receptors (AT2R), where AT1R stimulation induces vasoconstriction and remodeling, while AT2R stimulation promotes vasodilation and reduces remodeling. AT1R stimulation also results in the production and release of aldosterone. ACE2 converts Ang II into Ang-(1-7), activating the Mas receptor (MasR) and counteracting AT1R effects.

RAS activation is implicated in heart failure pathophysiology. It compensates for reduced cardiac output and is linked to cardiac hypertrophy, endothelial dysfunction, and oxidative stress [23,24,25,26,27]. The local formation of Ang II in the heart, depending on liver-derived angiotensinogen, induces hypertrophy even without high blood pressure, highlighting the role of local cardiac Ang II levels in hypertrophic response regulation [28,29,30]. This involves Ang II-induced synthesis and the release of TGFβ1, matrix metalloproteinases (MMPs), and matrix components like fibronectin and collagen [31,32,33], increasing myocardial stiffness and contributing to cardiac dysfunction [34,35]. Similar properties have been linked to aldosterone [36,37].

Therapies targeting the RAS, such as ACE inhibitors, have been explored in TAC-induced pressure overload models in mice. Current RAS-targeted treatments include renin inhibitors, ACE inhibitors (ACEis), and AT1R receptor blockers (ARBs), as shown in Figure 2. Renin inhibitors prevent renin from cleaving Ang I from angiotensinogen, while ACEis prevent the conversion of Ang I to Ang II, thereby simultaneously preventing AT2R activation. ARBs specifically block AT1R activation. Both ARBs and ACEis increase renin secretion by blocking Ang II feedback [38]. During ARB treatment, this results in elevated Ang II levels, which may now stimulate AT2R. In TAC mice, the ACEis captopril and ramipril have been shown to reduce post-TAC cardiac hypertrophy and fibrosis [39,40], whereas temocapril only reduced cardiac hypertrophy [41]. Enalapril treatment did not affect ejection fraction or heart weight in such models, but reduced plasma BNP levels, pulmonary edema, left atrial diameter, and overall mortality [42]. The deletion of ACE2 in mice exacerbates cardiac disease progression in pressure-overloaded hearts [43]. A recent study used the bacteria-derived ACE2-like carboxypeptidase B38-CAP in TAC mice to mimic ACE2 activity [44], converting Ang II to Ang-(1-7). This treatment decreased Ang II levels, attenuating hypertension, cardiomyocyte hypertrophy, and myocardial fibrosis compared to untreated TAC mice [44].

RAS inhibition is also commonly used in connective tissue disorders. The changes in ECM and signaling pathways that occur in these disorders are usually investigated through the use of mouse models. In Marfan syndrome, a lack of functional fibrillin-1, an important component of microfibrils, predisposes individuals to thoracic aortic aneurysms (TAA) and aortic dissections due to compromised ECM integrity [45,46]. Simultaneously, fibrillin-1 is a regulator of TGFβ1 [46]. Treatment with losartan, an AT1R antagonist, not only prevented development of TAA but also reduced deterioration of the aortic wall architecture and prevented abnormal aortic dilatation [46]. Treatment of Marfan syndrome does seem to rely on what mutations are present, as its effectiveness is greater in haploinsufficient mutations compared to dominant negative mutations [47]. Another ECM component important in elastic fiber formation is fibulin-4. Mice with a reduced expression of fibulin-4 develop TAA, accompanied by enlarged hearts [48]. Treatment with the renin inhibitor aliskiren reduced blood pressure without affecting survival [48]. In contrast, losartan reduced aneurysm size and left ventricular growth, improving aorta integrity, heart function, and survival. Since both inhibitors block the Ang II-AT1R pathway, the most likely explanation for this apparent discrepancy is that losartan additionally upregulates AT2R stimulation. Indeed, AT2R stimulation during AT1R blockade could be beneficial with regard to the prevention of TAA formation [48]. That being said, targeting the RAS is often hampered due to the concomitant rise in renin (resulting in increased angiotensin generation), implying that more than one RAS blocker might be required to enhance effectiveness.

### 3.2. Mineralocorticoid Receptor Antagonists

MRAs such as spironolactone and eplerenone have emerged as vital therapeutic agents in the management of cardiac pressure overload, particularly due to their well-documented anti-fibrotic properties. MRAs inhibit the actions of aldosterone, a hormone that contributes to cardiac fibrosis through its receptors located on cardiomyocytes and cardiac fibroblasts [36]. By blocking the mineralocorticoid receptor (MR), MRAs reduce sodium reabsorption and water retention, leading to decreased blood volume and reduced cardiac workload. This mechanism not only alleviates hypertension but also mitigates the detrimental effects of aldosterone-induced fibrosis in the myocardium. Activation of the MR also occurs on vascular smooth muscle cells [49]. Smooth muscle cell-specific MR knockout protects from adverse cardiac remodeling induced by pressure overload [50].

### 3.3. Beta Blockers

β-adrenergic receptors are G protein-coupled receptors targeted by noradrenaline and adrenaline [51]. Beta blockers inhibit β1 (occurring in the heart and kidney) and β2 (occurring in the heart and vessel wall) receptors. Selective β1 blockers such as atenolol and metoprolol are used to treat hypertension, and their mechanism of action likely involves the inhibition of renin release from the kidney. Non-selective beta blockers like propranolol also target the β2 receptor, which may result in unwanted vaso- and bronchoconstriction. Metoprolol, carvedilol, and bisoprolol are the preferred beta blockers to treat heart failure.

### 3.4. Sodium-Glucose Cotransporter 2 Inhibitors

The healthy heart is metabolically flexible, primarily deriving its energy from fatty acids. Changes in glucose consumption in response to stress may precede structural remodeling. SGLT2 is responsible for glucose reabsorption in the proximal tube of the kidney. Consequently, SGLT2is have been prescribed to type 2 diabetic patients to reduce glucose levels. Treating diabetes with SGLT2is also reduces mortality and the risk of adverse cardiovascular events by approximately 30%. This can be explained, in part, by how cardiomyocytes subjected to high glucose conditions increase SGLT2 expression [52]. In a rat model of type 1 diabetes mellitus, valsartan was complimented with the SGLT2i empagliflozin to study the dual blockade of Ang II-dependent hypertension and SGLT2 inhibition [2]. The dual treatment showed a synergistic effect on lowering blood pressure and reducing serum glucose levels and cardiac hypertrophy. Importantly, SGLT2i treatment reverted the increased expression of fibrotic markers in the kidney caused by valsartan treatment alone. Thus, RAS inhibition could benefit from combination therapy with SGLT2is.

Interestingly, the effectiveness of SGLTis seems to not be directly linked to the inhibition of SGLT2. SGLT2 knockout mice showed no cardioprotective effect during TAC-induced pressure overload [53], nor did SGLT2 deletion protect mouse hearts from reperfusion injury [54]. The current consensus thus suggests that the beneficial effects of SGLT2is in the heart concern off-target effects [55,56]. SGLT2 inhibitors also modulate the extracellular matrix by reducing fibrosis through mechanisms that involve decreased activation of pro-fibrotic signaling pathways, such as TGFβ1, which in turn limits the deposition of collagen and other ECM components, thereby mitigating adverse cardiac remodeling.

To uncover the link between cardiac remodeling and SGLT2i treatment, several studies have investigated the SGLT2i empagliflozin in mice with pressure overload without diabetes mellitus. Empagliflozin attenuated adverse left ventricular remodeling after TAC and reduced cardiac fibrosis [57]. This beneficial effect was attributed to empagliflozin’s direct binding to glucose transporters, improving mitochondrial oxidative phosphorylation, biogenesis, and morphology, thereby reducing ROS production, increasing autophagy, and attenuating apoptosis in cardiomyocytes [58]. To what degree SGLT2 truly occurs in the heart is still being debated [2,54,56].

The SGLTi dapagliflozin also behaves in an SGLT2-independent manner. It alleviated cardiac hypertrophy and increased ejection fraction in TAC mice without diabetes [59], while simultaneously inhibiting myocardial fibrosis and cardiomyocyte apoptosis through p38 and JNK phosphorylation inhibition [60]. Studies in SLGT2 knockout mice subjected to TAC showed that dapagliflozin can inhibit macrophage inflammation and subsequent cardiac fibroblast activation [55]. Ertugliflozin reduced cardiac insulin and increased AMPK signaling, resulting in reduced cardiac mTOR activation, a pathway relevant in cardiac hypertrophy and adverse cardiac remodeling [61]. Sotagliflozin, a dual SGLT1/2 inhibitor, also reduced cardiac hypertrophy and fibrosis [62].

These studies highlight a connection between SGLT2i use and cardiac remodeling through reduced ROS formation and mitochondrial stress and the prevention of cardiac fibroblast activation and the inflammation response of immune cells. An important side effect of SGLT2i treatment is that it can increase renin levels [2].

## 4. Novel Therapies Targeting Cardiac ECM Remodeling

While the previously mentioned therapies are now incorporated into the clinical guidelines for heart failure, there is still a need for further improvement. Specifically, targeting ECM-related components could prove vital in combination with already existing therapies. Figure 3 summarizes recent developments that may help to find new drug targets to counteract ECM remodeling during cardiac pressure overload.

### 4.1. Transforming Growth Factor β1 Signaling

One approach to prevent pathological remodeling of the heart is to inhibit cardiac fibrosis. TGFβ1 signaling is one of the most widely investigated factors in fibrosis. TGFβ1 is expressed and sequestered into the surrounding ECM as a large latent complex [45]. Upon release from the ECM, TGFβ1 activates cardiac fibroblasts, leading to remodeling. Imbalances in TGFβ1 signaling have been associated with increased inflammation and fibrosis, leading to heart failure [63]. TGFβ1 forms an integral part of the cardiac maladaptive response upon pressure overload due to its pro-fibrotic properties [64]. Mice overexpressing TGFβ1 displayed a ten-fold increase in myocardial TGFβ1 levels, without significant alterations in MMP-2 and MMP-9 concentrations [65]. This finding suggests that the pro-fibrotic activity of TGFβ1 is independent of MMP/TIMP activity. TGFβ1 overexpression precedes the increase of collagen types I and III and decreases protein degradation by inhibiting collagenase expression [65]. Recently, the compound Si-Miao-Yong-An decoction (SMYAD), an inhibitor of the TGFβ1/Smad and TGFβ1/TAK1/p38 signaling pathways, was administered to TAC mice to assess its effects on cardiovascular disease development [66]. Interestingly, SMYAD treatment in TAC mice led to significant improvement in cardiac fibrosis and function [66]. Furthermore, circulating TGFβ1 plasma levels in both TAC mice and patients with aortic stenosis correlated with the accumulation of cardiac ECM components, including collagen types I, III, and fibronectin, as well as with myocyte hypertrophy [67].

The TGFβ1/Smad2 signaling pathway induces cardiac hypertrophy, ventricular remodeling, and myocardial fibrosis [65,66]. Geniposide (GE), a major iridoid in gardenia fruit extract, suppressed TGFβ1/Smad2 signaling, reducing inflammation and ventricular remodeling [68,69]. Another player in TGFβ1/Smad2 signaling is rho-associated kinase-2 (ROCK2). ROCK2 mediates cardiac fibrosis by interacting with the TGFβ1/Smad2 signaling pathway. Inhibition of ROCK2 with belumosudil, a ROCK2-specific inhibitor, reduced cardiac fibroblast activation and proliferation induced by TGFβ1. Both compounds reduced cardiac hypertrophy and fibrosis induced by TAC. Next to Smad2, Smad3 is similarly responsible for TGFβ1 signaling. Upon phosphorylation, Smad3 and Smad2 form a complex with Smad4 before translocating to the nucleus, regulating fibroblast function [70]. Pirfenidone, an anti-pulmonary fibrosis medicine, has been shown to inhibit vimentin and α-SMA expression, an important factor in fibroblast-to-myofibroblast transition in response to TGFβ1 [71]. In TAC mice, pirfenidone reduced both cardiac expression of TGFβ1 as well as phosphorylation of Smad3, resulting in less cardiac fibroblast proliferation [72].

Nintedanib, a tyrosine kinase inhibitor used to treat idiopathic pulmonary fibrosis, mitigated cardiac fibrosis in TAC mice by preventing myofibroblast transformation [73]. Nintedanib not only inhibited TGFβ1-induced SMAD3 phosphorylation and the production of fibrogenic proteins, but was also able to reduce myocardial and systemic inflammation [73]. These effects persisted even after treatment cessation in these mice [73].

TGFβ1, due to its prominent role in myofibroblast activation as well as inflammation, has therefore shown promising results in the treatment of remodeling. Targeted suppression of TGFβ1 results in a reduced adaptive response, potentially benefitting as a combination treatment with hypertensive drugs.

### 4.2. Secreted Protein Acidic and Rich in Cysteine (SPARC)

Secreted Protein Acidic and Rich in Cysteine (SPARC) regulates extracellular matrix composition and remodeling by interacting with matrix proteins and influencing their deposition, which affects tissue structure and cellular behavior. In the heart, it is expressed primarily in fibroblasts, but it is also present in endothelial cells and cardiomyocytes [74,75]. Bone marrow-derived cells have also been discovered as cellular origins of cardiac SPARC production [76]. SPARC has shown involvement in various cellular mechanisms, ranging from cell migration and proliferation to MMP activity [77]. SPARC is perceived as a stabilizer of pre-existent fibrillar collagen, leading to diastolic rigidity by means of cross-linking collagen fibrils, resulting in diastolic dysfunction [78].

SPARC contributes to a maladaptive cardiac remodeling response upon pressure overload. Fibroblast transdifferentiation into collagen-producing myofibroblasts is controlled by SPARC [79]. Cardiac-pressure-overloaded TAC mice showed an increased expression of SPARC, fibrillary proteins, and soluble and insoluble collagens, contributing to an increased left ventricular weight [78]. SPARC proteins modulate collagen metabolism in the normal and pressure-overloaded myocardium by post-translational processing [78]. In TAC mice lacking SPARC, myocardial interstitial insoluble collagen levels were decreased compared to WT TAC mice [78]. The level of novel collagen production in mice lacking SPARC was similar to WT mice post-TAC, suggesting that SPARC proteins are a post-translational stabilizer of collagen fibers [80]. SPARC knockout mice preserved their left ventricular weight after TAC [78]. Additionally, mice lacking SPARC demonstrated significantly attenuated fibrillar collagen deposition and diastolic dysfunction [78]. The level of novel collagen production was similar to WT mice without pressure overload [80]. A lack of SPARC thus resulted in less stabilization of fibrillar collagen. Similarly, wild-type mice transplanted with SPARC knockout bone marrow showed an increased papillary muscle stiffness after pressure overload, while SPARC knockout mice with wild-type bone marrow showed no increase in stiffness [76]. Drugs which downregulate SPARC activity may therefore be beneficial in preventing fibrotic remodeling.

### 4.3. Matrix Metalloproteinases and Tissue Inhibitors of Matrix Metalloproteinases

Matrix metalloproteinases (MMPs) are zinc-dependent endopeptidases that regulate ECM turnover by degrading various ECM proteins [81]. MMPs play a pivotal role by modulating ECM composition and promoting fibrosis in response to pressure overload, as previously described in mammals ranging from mice to humans [81]. Identified MMPs in the human myocardium consist of collagenases (MMP-1 and MMP-13), gelatinases (MMP-2 and MMP-9), stromelysins (MMP-3), and membrane-type MMPs (MMP-14) [81]. MMP-2, or gelatinase A, is a matricellular protein present within the mammalian myocardium, of which ECM-collagens, gelatins, and components of the basement membrane form its substrates [81]. MMP-2 protein levels increase in the mouse myocardium following pressure overload, as well as its gelatinolytic activity [82]. In an MMP-2 homozygous knockout (KO) mouse model system, TAC resulted in a significantly lower-end diastolic pressure and left ventricular (LV) weight compared to wild-type (WT) mice [82]. Furthermore, MMP-2 knockout mice demonstrated attenuated interstitial fibrosis and increased myocyte hypertrophy, supporting a role for MMP-2 in cardiac remodeling [82].

MMP-9, or gelatinase B, also contributes to cardiac remodeling [81]. The stress-inducible transcriptional regulator p8, required for MMP-9 expression, is upregulated in mice upon TAC [83]. In contrast, p8 knockout mice showed a reduced expression of MMP-9 compared to wild-type mice. In response to a reduced expression of MMP-9, mice lacking p8 showed lower levels of myocardial fibrosis after TAC [83]. Recent studies have shown that increased interleukin 4 (IL-4) derived from macrophages increases both MMP-9 mRNA and protein levels, leading to an elevated ECM degrading activity similar to MMP-2 [84].

TIMPs regulate the enzymatic activity of MMPs [81]. The protein expression levels of TIMP-2, in particular, significantly increase following pressure overload [85]. TIMP-2 is essential for the integrity of cardiomyocyte–ECM interaction via ECM-adhesion molecules, comprised of laminin and fibronectin, and by conserving a sustained level of integrinβ1D protein levels [80]. ECM remodeling in TIMP-2 deficient mice is, most importantly, impaired by the formation of fibrillar protein depots [80]. The pro-fibrotic tendency of the cardiac tissue in TIMP-2 deficient mice can be attributed to an increase in the SPARC protein level [80]. TIMP-4, like TIMP-2, is upregulated upon TAC-induced pressure overload [86]. TIMP-4 deficiency in TAC mice results in an increase in left ventricular hypertrophy, ventricular dilatation, and ultimately cardiac dysfunction in comparison to WT TAC mice [86]. In TIMP-4-overexpressing mice, the degree of myocyte hypertrophy was attenuated [86]. TIMP-4 KO mice, in contrast, demonstrated a significant increase in myocardial collagen type I and III expression following pressure overload compared to TIMP-4-overexpressing mice and wild-type mice [86].

MMPs/TIMPs demonstrate opposite effects within the context of cardiac remodeling. TIMPs exhibit cardio-protective properties, whilst MMPs contribute to a maladaptive cardiac remodeling response by stimulating cardiac fibrosis and dysfunction. It stands to reason that stimulating TIMP-2 and TIMP4 activity while inhibiting MMP-2 and MMP-9 could tackle remodeling early in cardiac disease development.

### 4.4. Thrombospondins

Thrombospondins (TSPs) are matricellular proteins that exhibit complex functions. Currently, five members of the TSP family have been described. TSP-1 can interact with growth factors and regulators, proteases, receptors (commonly integrins), and ECM proteins such as collagens and proteoglycans [87]. TSP-1 and -2 are involved in wound healing as well as pressure overload [88]. Specifically, TSP-1 is upregulated in TAC-induced pressure overload [89]. TSP-1-deficient mice demonstrate increased early hypertrophy by a rapid increase in LV weight due to myofibroblast expansion, facilitating the enhancement of ventricular dilatation [89]. TSP-1 is considered to be required for the prevention of LV dilatation in order to maintain cardiac function [89]. Furthermore, in the absence of TSP-1, cardiomyocytes demonstrate exacerbated loss in sarcomeres [89]. TSP-4 expression is, similar to TSP-1, stimulated in response to pressure overload [90]. The heart weight of TSP-4 KO mice was significantly higher compared to WT mice post-TAC [64]. The increase in cardiac weight is attributed to an accumulation of interstitial collagen in the ECM [64]. In the absence of TSP-4, the expression of TGFβ2 was simultaneously decreased [64]. The previous findings are indicative of the mechanism of action regarding the pro-fibrotic activity of TSP-4 as part of a maladaptive remodeling response [64]. These findings indicate that TSPs exhibit cardio-protective properties in terms of cardiac dilatation and function, concurrently demonstrating pro-fibrotic activity by stimulating the TGFβ signaling pathway.

### 4.5. Protein Glycosylation

The cardiovascular system, like many other parts of the body, is controlled in part by protein glycosylation. The cardiac and vascular walls are lined with receptors that are *N*- and *O*-glycosylated. Intracellular signaling pathways also rely on glycosylation. Beta-1,4-galactosyltransferase V (B4GALT5), involved in protein glycosylation, is upregulated in models of TAC-induced pressure overload [91]. Downregulation of B4GALT5 alleviated fibrosis after TAC, while overexpression aggravated fibrosis formation in TAC mice [91]. B4GALT5 regulates cardiac fibroblast activation through the Akt/GSK-3β/β-catenin signaling pathway by inhibiting β-catenin ubiquitination [91]. B4GALT5 also elevates expression of and interacts with lumican, a small proteoglycan localized in the ECM. Lumican expression increases during heart failure and cardiac fibrosis, and the stimulation of lumican was shown to further increase cardiac fibrosis [92,93]. Thus, inhibition of B4GALT5 may be a complementary target to prevent cardiac fibrosis.

One subtype of *O*-glycosylation is *O*-GlcNAcylation. *O*-GlcNAc transferase (OGT) has been investigated in cardiac remodeling. OGT adds *O*-linked β-N-acetylglucosamine to the serine/threonine residues of proteins and is upregulated during cardiac hypertrophy and subsequent heart failure [94]. *O*-GlcNAcylation also increases in diabetic patients. GSK-3β induces cardiac hypertrophy by promoting NFAT nuclear translocation [94]. GSK-3β activation through OGT inhibits compensatory hypertrophy, potentially increasing susceptibility to pressure overload [94]. In OGT-overexpressing transgenic mice, cardiac dysfunction was observed without significantly increased cardiac hypertrophy after TAC [94]. Treatment with the GSK-3β inhibitor TDCD-8 reversed the inhibitory effect on cardiac hypertrophy in these mice [94]. This suggests that OGT is involved not only in cardiac hypertrophy but also in fibrosis and heart failure progression.

Lastly, CD147 is a transmembrane glycoprotein receptor that activates matrix metalloproteinases and promotes inflammation. CD147 regulation primarily depends on glycosylation in the heart and is elevated in response to TAC [95,96]. CD147 induces cardiac remodeling through TRAF2-TAK1 signaling [96]. Blocking CD147glycosylation could therefore be a potential target for the prevention of maladaptive cardiac remodeling.

Protein glycosylation is very much involved in cardiac pressure overload. Yet, little is known about the side effects of systemic inhibition of these processes, and while further research is needed, targeted organ-specific inhibition of the abovementioned glycosylation processes could improve the disease outcome.

## 5. Conclusions and Future Perspectives

Heart failure attributable to pressure overload presents an urgent health problem worldwide. The burden of this disease on the patient and, as a consequence, on the healthcare system, is of great magnitude and significance and therefore in need of novel insights into the pathophysiological mechanisms of pressure overload-induced heart failure. Identified cardiac ECM proteins are demonstrably involved in maladaptive remodeling in response to TAC-induced pressure overload by modulating ECM assembly, composition, and metabolism. The remodeled cardiac ECM is a key orchestrator of pro-fibrotic activity, leading to cardiac fibrosis and eventual cardiac dysfunction. Identifying the involved ECM proteins therefore serves as a stepping stone towards the discovery of potential novel therapeutic targets which can counteract the pro-fibrotic tendencies and therefore prevent the development of heart failure. Pharmacological interventions specifically targeting the ECM may allow long-term recovery following pressure overload. Furthermore, understanding the relationship between cardiac remodeling and aortic diseases can provide a comprehensive approach to treating both conditions. Combining current RAS-targeting drugs with new therapies aimed at ECM remodeling could improve long-term cardiovascular outcomes.

## Figures and Tables

**Figure 1 cells-13-01685-f001:**
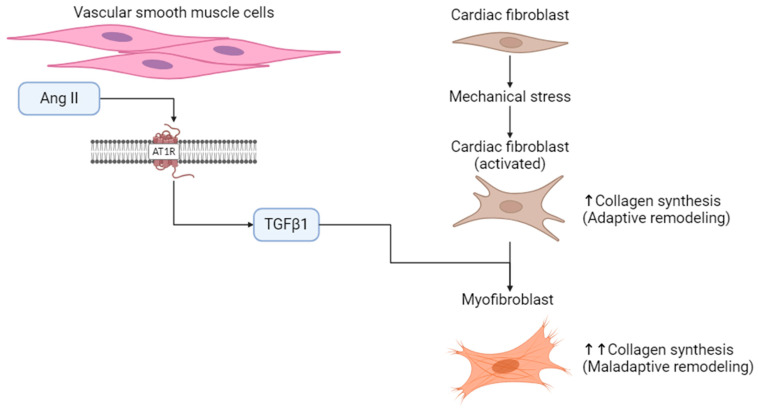
Example image showing the molecular response of vascular smooth muscle cells to Ang II. AT1R activation can potentially trigger maladaptive remodeling of the ECM through TGFβ1 activation, leading to collagen synthesis and maladaptive remodeling. Upward arrows indicate an increase in collagen synthesis. Created with BioRender.com.

**Figure 2 cells-13-01685-f002:**
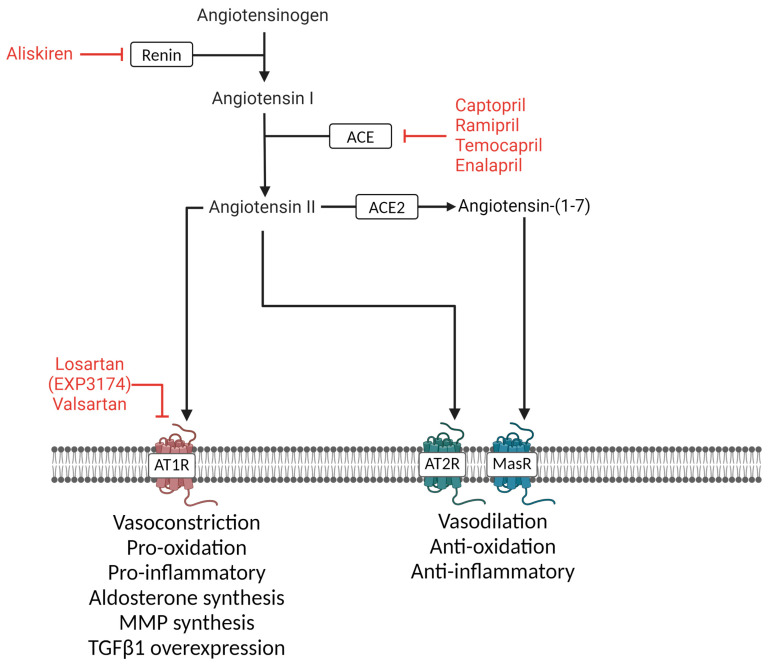
Drug targets in RAS and their implications. The RAS is depicted in black. Example RAS inhibitors and treatments are shown in red. The downstream effects of each receptor are written underneath. Created with BioRender.com.

**Figure 3 cells-13-01685-f003:**
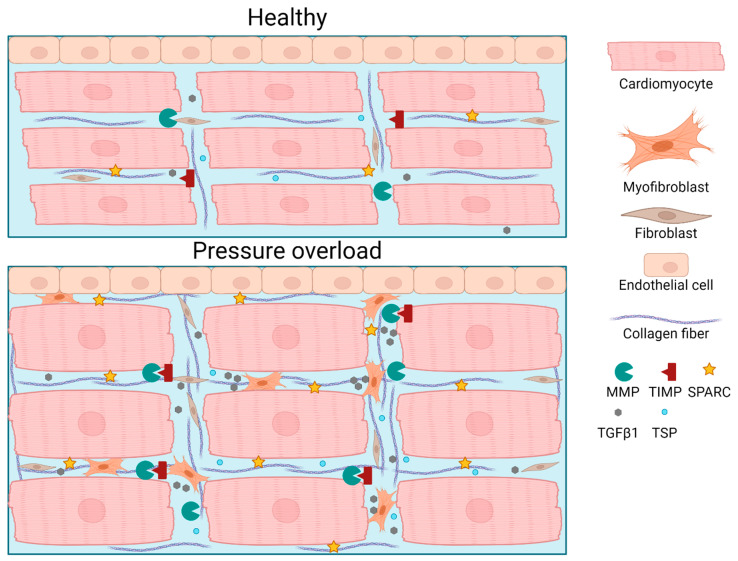
Schematic overview of maladaptive ECM remodeling after pressure overload compared to the healthy situation. Increased excrection of the ECM interacting proteins, MMPs, and TGFβ1 results in an increase in myofibroblast proliferation and collagen deposition, as well as a hypertrophic response in cardiomyocytes. Created with BioRender.com.

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
