# Peer review of "The Extracellular Matrix and Cardiac Pressure Overload: Focus on Novel Treatment Targets"

_cells, 2024, doi:10.3390/cells13201685_

Round 1

Reviewer 1 Report

Comments and Suggestions for Authors

The manuscript consists of a review of current therapies for the treatment of cardiac fibrosis and other changes that occur in the heart as a result of pressure overload. It also delves into cellular mechanisms that may be potential targets for experimental treatments

The paper presents an interesting and original approach

Have only few comments

There is no mention of MRA antagonists. Since anti-fibrotic proprieties of these drugs have been widely demonstrated  at cardiac level, in my opinion authors should add a paragraph dedicated to  MRA antagonists-

SGLT-2 cannot be considered “new drugs” although, it must be admitted that the spectrum of beneficial actions performed by this group of drugs is continually expanding. Since SGTL2 are currently used in the entire population of HF regardless of the value of the ejection fraction and are strongly recommended by guidelines they can hardly be included  in the paragraph “Novel drug treatments targeting cardiac ECM remodeling”. rather, in my opinion the  aragraph  dedicated  to SGLT2 should follow that on beta blockers.

Author Response

Reviewer #1:

The manuscript consists of a review of current therapies for the treatment of cardiac fibrosis and other changes that occur in the heart as a result of pressure overload. It also delves into cellular mechanisms that may be potential targets for experimental treatments

The paper presents an interesting and original approach

Have only few comments

Comment 1: There is no mention of MRA antagonists. Since anti-fibrotic proprieties of these drugs have been widely demonstrated at cardiac level, in my opinion authors should add a paragraph dedicated to MRA antagonists.

Response 1: Thank you for your insightful comment regarding the omission of mineralocorticoid receptor antagonists (MRAs) in our discussion of current therapies for cardiac pressure overload. We agree that MRAs, such as spironolactone and eplerenone, have demonstrated significant anti-fibrotic properties at the cardiac level and are critical to consider in this context.

We added a dedicated paragraph to highlight the role of MRAs in cardiac remodeling and their mechanisms of action. Specifically, we discuss how activation of the receptor occurs not only in smooth muscle cells, but also in cardiomyocytes and fibroblasts, and that smooth muscle cell-specific knockout of MR protects against pressure overload-induced remodeling. This can be found on page 4, paragraph 1, line 112 of the revised manuscript.

Comment 2: SGLT-2 cannot be considered “new drugs” although, it must be admitted that the spectrum of beneficial actions performed by this group of drugs is continually expanding. Since SGTL2 are currently used in the entire population of HF regardless of the value of the ejection fraction and are strongly recommended by guidelines they can hardly be included  in the paragraph “Novel drug treatments targeting cardiac ECM remodeling”. rather, in my opinion the paragraph dedicated to SGLT2 should follow that on beta blockers.

Response 2: Thank you for pointing this out. We agree with this comment. Therefore, we have updated the manuscript and moved the section about SGLT-2 to follow beta blockers. This can be found on page 5, paragraph 4, line 159 of the revised manuscript.

Reviewer 2 Report

Comments and Suggestions for Authors

In this interesting review, the Authors described all current therapies for treating pressure overload. They also explored the ECM as therapeutic target.

Traditional and novel therapeutic targets might counteract the pro-fibrotic tendencies and therefore prevent the development of heart failure, secondary to pressure-overload.

The manuscript is well written and full of interesting information.

I have only one suggestion for the Authors.

Given that myocardial fibrosis is the main responsible for the subsequent occurrence of heart failure, on line 170, paragraph "Cardiac ECM remodeling in response to pressure overload", after the reference number 47, the Authors could also state that the ECM remodeling in response to pressure overload may be detected, in the clinical practice, by using strain echocardiographic imaging. This innovative imaging modality allows the cardiologists to detect a subclinical impairment in myocardial strain parameters before the occurrence of left ventricular systolic dysfunction, expressed by left ventricular ejection fraction (LVEF) depression. The lower is the magnitude of left ventricular global longitudinal strain (LV-GLS), assessed by speckle tracking echocardiography, the greater is the collagen deposition in the myocardial tissue. LV-GLS is strongly correlated with myocardial fibrosis.

With this regard, the Authors could cite the following articles:  PMID: 30448148, PMID: 23701925 and PMID: 36142856.

Author Response

Reviewer #2:

In this interesting review, the Authors described all current therapies for treating pressure overload. They also explored the ECM as therapeutic target.

Traditional and novel therapeutic targets might counteract the pro-fibrotic tendencies and therefore prevent the development of heart failure, secondary to pressure-overload.

The manuscript is well written and full of interesting information.

I have only one suggestion for the Authors.

Comment 1: Given that myocardial fibrosis is the main responsible for the subsequent occurrence of heart failure, on line 170, paragraph "Cardiac ECM remodeling in response to pressure overload", after the reference number 47, the Authors could also state that the ECM remodeling in response to pressure overload may be detected, in the clinical practice, by using strain echocardiographic imaging. This innovative imaging modality allows the cardiologists to detect a subclinical impairment in myocardial strain parameters before the occurrence of left ventricular systolic dysfunction, expressed by left ventricular ejection fraction (LVEF) depression. The lower is the magnitude of left ventricular global longitudinal strain (LV-GLS), assessed by speckle tracking echocardiography, the greater is the collagen deposition in the myocardial tissue. LV-GLS is strongly correlated with myocardial fibrosis. With this regard, the Authors could cite the following articles:  PMID: 30448148, PMID: 23701925 and PMID: 36142856.

Response 1: Thank you for your insightful feedback regarding the detection of myocardial fibrosis and the potential inclusion of speckle tracking echocardiography (STE) in our discussion. We appreciate your suggestion to elaborate on the clinical relevance of STE in identifying subclinical myocardial strain impairments. we have revised the manuscript to include a more detailed explanation of how STE can detect fibrotic before the onset of left ventricular systolic dysfunction. This can be found on page 8, paragraph 2, line 283 of the revised manuscript.

Reviewer 3 Report

Comments and Suggestions for Authors

The Authors open their contribution by focusing on TAC in mice then drive to heart failure and trearments in humans and finish by reviewing molecular mechanisms leading to fibrosis and highly experimental interventions to modulate the hypertrophic heart and the underlying structural alterations. Overall, the text raises an impression of an erratic and confused document possibly related to a badly edited compilation of the different contributions of the individual panel of Authors of the document.

In addition there are several points, mainly related to misquoted references, which, in my opinion have to be carefully avoided in a paper aimed at reviewing evidence on a specific topic.   

Lines 46-48: "transverse aortic constriction (TAC)... mimics pressure overload by reducing the diameter of the mouse aorta... This  significant  reduction  results  in  up  to  70%  less  blood  flow  to  the  left ventricle..." Less blood flow to the left ventricle? Please clarify

Lines 109-123: "Calcium channel blockers etc" : meaningless paragraph in the context of the review because of lack of pertinent evidence on this class of drugs and confusion between the different meachanism of action of dyhydropriridines and phenylalkilamines

Lines 128-129 " β1  blockers  such  as  nebivolol,  atenolol,  and  metoprolol are used to treat hypertension by inhibiting renin release in the kidney [22]."Reference 22 misquoted and used to draw conclusions not pertinent to the quoted document

Line 134 " Long term effects of pressure overload include physical changes..." Physical? please clarify

Lines 135-136: " Combination  therapy  may  offer  greater  efficacy than single-drug treatments, but may also increase risks, such as hypotension [23, 24]." Again: ref 23 and 24 misquoted and used to draw unsupported conclusions

Lines 137-138: "Moreover, most drugs alleviate cardiac disease symptoms without addressing the under lying cause" What drugs? what symptoms? Besides, I understood from the opening paragraph that the paper was primarily dealing with TAC models of cardiac hypertrophy in experimental animals

Lines 304-305: "Combination therapy with renin inhibitors such as aliskiren may be required, as SGLT2i have been shown to  increase renin levels (74)" Again: Ref 74 deals with valsartan and, although reasonable as a choice for inhibiting renin, aliskiren is not even mentioned in the paper

Author Response

Reviewer #3:

The Authors open their contribution by focusing on TAC in mice then drive to heart failure and treatments in humans and finish by reviewing molecular mechanisms leading to fibrosis and highly experimental interventions to modulate the hypertrophic heart and the underlying structural alterations. Overall, the text raises an impression of an erratic and confused document possibly related to a badly edited compilation of the different contributions of the individual panel of Authors of the document.

In addition there are several points, mainly related to misquoted references, which, in my opinion have to be carefully avoided in a paper aimed at reviewing evidence on a specific topic.

Comment 1: Lines 46-48: "transverse aortic constriction (TAC)... mimics pressure overload by reducing the diameter of the mouse aorta... This  significant  reduction  results  in  up  to  70%  less  blood  flow  to  the  left ventricle..." Less blood flow to the left ventricle? Please clarify

Response 1: Thank you for pointing this out. We have corrected the sentence to “This significant constriction results in up to 70% reduction of left ventricular outflow, depending on the resulting diameter, leading to cardiac pressure overload and subsequent left ventricular hypertrophy and eventually heart failure.” We further have moved the section regarding the mouse models to later in the manuscript. This correction can be found in the revised manuscript on page 9, paragraph 2, line 320.

Comment 2: Lines 109-123: "Calcium channel blockers etc" : meaningless paragraph in the context of the review because of lack of pertinent evidence on this class of drugs and confusion between the different mechanism of action of dyhydropriridines and phenylalkilamines.

Response 2: Thank you for pointing this out. We agree that this paragraph contributes little in the context of our manuscript, as these are mainly blood pressure lowering drugs. We have decided to remove it from the revised manuscript.

Comment 3: Lines 128-129 " β1  blockers  such  as  nebivolol,  atenolol,  and  metoprolol are used to treat hypertension by inhibiting renin release in the kidney [22]."Reference 22 misquoted and used to draw conclusions not pertinent to the quoted document.

Response 3: Thank you for pointing this out. We have now rewritten this paragraph, and have omitted the use of reference 22 in this sentence, also because this is common knowledge. (Page 5, paragraph 2, line 142)

Comment 4: Line 134 " Long term effects of pressure overload include physical changes..." Physical? please clarify.

Response 4: Thank you for pointing this out. We have corrected the word physical and changed it to “maladaptive morphological changes”. This can be found in the revised manuscript on page 7, paragraph 2, line 238.

Comment 5: Lines 135-136: " Combination  therapy  may  offer  greater  efficacy than single-drug treatments, but may also increase risks, such as hypotension [23, 24]." Again: ref 23 and 24 misquoted and used to draw unsupported conclusions.

Response 5: We agree with the comment, and have rephrased the sentence to how it is intended: “Combination therapy may offer greater efficacy than single-drug treatments, but care should be taken to avoid increased risk of side-effects” and we have removed the references. This can be found on page 7, paragraph 2, line 239 of the revised document.

Comment 6: Lines 137-138: "Moreover, most drugs alleviate cardiac disease symptoms without addressing the under lying cause" What drugs? what symptoms? Besides, I understood from the opening paragraph that the paper was primarily dealing with TAC models of cardiac hypertrophy in experimental animals.

Response 6: We agree that this sentence is unclear and have removed it.

Comment 7: Lines 304-305: "Combination therapy with renin inhibitors such as aliskiren may be required, as SGLT2i have been shown to  increase renin levels (74)" Again: Ref 74 deals with valsartan and, although reasonable as a choice for inhibiting renin, aliskiren is not even mentioned in the paper.

Response 7: Thank you for pointing this out. We agree with this comment and have removed this sentence.

Round 2

Reviewer 3 Report

Comments and Suggestions for Authors

The revised version did not answer my main criticism of this paper i.e. "The Authors open their contribution by focusing on TAC in mice then drive to heart failure and treatments in humans and finish by reviewing molecular mechanisms leading to fibrosis and highly experimental interventions to modulate the hypertrophic heart and the underlying structural alterations. Overall, the text raises an impression of an erratic and confused document possibly related to a badly edited compilation of the different contributions of the individual panel of Authors of the document."

By the way, the text as a whole is not coherent with the abstract session of the paper "A critical aspect of cardiac disease is the remodeling of the heart, where compensatory changes in the extracellular matrix exacerbate disease progression. This review explores the processes and changes occurring in pressure overloaded hearts with respect to the extracellular matrix, focusing on recent advances in targeted therapies for maladaptive cardiac remodeling in transverse aortic constriction-induced pressure overload models."

Author Response

Comment 1: The revised version did not answer my main criticism of this paper i.e. "The Authors open their contribution by focusing on TAC in mice then drive to heart failure and treatments in humans and finish by reviewing molecular mechanisms leading to fibrosis and highly experimental interventions to modulate the hypertrophic heart and the underlying structural alterations. Overall, the text raises an impression of an erratic and confused document possibly related to a badly edited compilation of the different contributions of the individual panel of Authors of the document."

Response 1: We regret that our revision did not answer the main criticism of the reviewer. We have further revised the following points:

  1. We have now re-ordered the sections to obtain a more logical structure; we start by explaining the current challenges in halting disease progression in the pressure overloaded heart. We explain what models are used to study the systems involved. We then proceed to summarize the recent therapies in treating pressure overload, and end with the novel targets of the ECM.
  2. We have altered sentences to improve the cohesion between sections.

Comment 2: By the way, the text as a whole is not coherent with the abstract session of the paper "A critical aspect of cardiac disease is the remodeling of the heart, where compensatory changes in the extracellular matrix exacerbate disease progression. This review explores the processes and changes occurring in pressure overloaded hearts with respect to the extracellular matrix, focusing on recent advances in targeted therapies for maladaptive cardiac remodeling in transverse aortic constriction-induced pressure overload models."

Response 2: We agree with the comment and we have revised the abstract to better align with the (revised) main text: "A critical aspect of cardiac disease is the remodeling of the heart, where compensatory changes in the extracellular matrix exacerbate disease progression. This review explores the processes and changes occurring in the pressure-overloaded heart with respect to the extracellular matrix. It further summarizes current treatment strategies, and then focuses on novel treatment targets for maladaptive cardiac remodeling, derived from transverse aortic constriction-induced pressure overload animal models."